# Unequal expression: Social position modulates APOE genotype risk of dementia

José M. Aravena[1,2,3*], Xi Chen[4,5], Becca R. Levy[1,5,6]

1 Department of Social & Behavioral Sciences, School of Public Health, Yale University, New Haven, Connecticut, United States of America, 2 Instituto de Nutrición y Tecnología de los Alimentos (INTA), Universidad de Chile, Santiago, Chile, 3 Facultad de Medicina Clínica Alemana, Universidad del Desarrollo, Santiago, Chile, 4 Department of Health Policy and Management, School of Public Health, Yale University, New Haven, Connecticut, United States of America, 5 Institution for Social and Policy Studies, Yale University, New Haven, Connecticut, United States of America, 6 Department of Psychology, Yale University, New Haven, Connecticut, United States of America

* jose.aravena@yale.edu; jm.aravenac@gmail.com

## Abstract

### Background

Although social position plays a pivotal role in cognitive aging, most dementia prevention strategies and risk prediction models continue to emphasize biomedical and genetic factors (particularly APOE status). This disconnect raises critical questions about how social environments may shape the effect of genetic risk on dementia. We examined how APOE alleles interact with social adversity to determine dementia risk.

### Methods

We conducted an observational analysis using two large cohort studies—the Health and Retirement Study (HRS) and the English Longitudinal Study of Ageing (ELSA)—including individuals aged 55 years or older without dementia at baseline. A social adversity index was constructed based on the five domains of social determinants of health outlined in the *Healthy People 2030* framework: education access, economic stability, healthcare quality, neighborhood environment, and social context. Participants were classified as having low (APOE-ε2), intermediate (APOE-ε3/ε3), or high (APOE-ε4) genetic risk of dementia. Dementia was ascertained via clinical diagnosis, cognitive testing, or validated caregiver report. Cox proportional hazards models were used in each cohort, and estimates were pooled using random-effects adjusting for covariates.

### Results

A total of 9,849 participants (HRS = 5,797; ELSA = 4,052) were followed for up to 12 years. Genetic effects were most pronounced among individuals with social advantage (reference: APOE-ε3/ε3 with social advantage; APOE-ε2 HR = 0.67,

**Data availability statement:** The data used in this study come from two publicly available, third-party cohort studies: the Health and Retirement Study (HRS) and the English Longitudinal Study of Ageing (ELSA). Health and Retirement Study (HRS) data are available to qualified researchers from the Health and Retirement Study (HRS) website (https://hrs.isr.umich.edu/data-products). Access to core and phenotypic data requires user registration. Access to genetic data requires an approved restricted data application, Institutional Review Board (IRB) approval, and a signed Data Use Agreement. Further details and application instructions for genetic data are available at https://hrs.isr.umich.edu/data-products/genetic-data. For assistance, contact Health and Retirement Study (HRS) via email (hrsquestions@umich.edu). English Longitudinal Study of Ageing (ELSA) data are available through the UK Data Service website (https://discover.ukdataservice.ac.uk/series/?sn=200011). Registration is required. Core datasets are available under the End User License. Genetic data require a separate application, submitted through the English Longitudinal Study of Ageing (ELSA) website (https://www.elsa-project.ac.uk/genetic-data-access). Support is available by contacting the UK Data Service via email (help@ukdataservice.ac.uk). Interested researchers can replicate the study findings in their entirety by obtaining access to the HRS and ELSA datasets through the procedures described above and following the methodology provided in the paper and its Supporting Information files. The authors did not have any special access privileges. All data were obtained using publicly available procedures, and no proprietary or restricted-access variables were used beyond what is available to other qualified researchers.

**Funding:** The Health and Retirement Study (HRS) is supported by the National Institute on Aging (NIA U01AG009740) and the Social Security Administration funding the Health and Retirement Study. The English Longitudinal Study of Ageing (ELSA) is funded by the National Institute on Aging (Ref: R01AG017644) and by a consortium of UK government departments: Department for Health and Social Care; Department for Transport; Department for Work and Pensions, which is coordinated by the National

95%CI = 0.48–0.93; APOE-ε4 HR = 1.68, 95%CI = 1.37–2.06). In contrast, those experiencing high social adversity had elevated dementia risk regardless of genotype (reference: APOE-ε3/ε3 with social advantage; APOE-ε2 HR = 3.26, 95%CI = 2.06–5.16; APOE-ε3/ε3 HR = 3.12, 95%CI = 2.47–3.95; APOE-ε4 HR = 3.21, 95%CI = 2.34–4.41). Notably, individuals with high genetic risk but social advantage had a lower dementia risk than those with low genetic risk but high social adversity.

## Conclusions

The influence of genetic risk on dementia appears shaped by social position. Addressing social adversity may reduce dementia risk across genotypes and enhance equity in dementia prevention strategies.

## Introduction

Dementia is a growing global public health challenge [1]. In response, prevention science and risk prediction strategies have increasingly focused on identifying biomedical and genetic risk factors to inform early intervention. Among genetic markers, the apolipoprotein E (APOE) gene stands out as the most robust predictor of late-onset Alzheimer's disease (AD) and vascular dementia risk [2]. These insights have led to the increasing use of polygenic risk scores, biomarker testing, and genotype-stratified interventions aimed at identifying individuals at high risk [3,4].

However, this precision approach often overlooks the profound influence of social determinants of health, which shape cognitive aging through lifelong exposure to structural disadvantage [5]. Social adversity, defined as cumulative disadvantage across domains such as education, income, healthcare access, neighborhood, and social environment, [6] has emerged as a critical driver of dementia risk [7]. Yet, prevention strategies and risk prediction models frequently incorporate social factors merely as covariates or isolated determinants of disease, rather than recognizing them as broader contextual forces that can fundamentally shape how individual-level risks, such as genetic predisposition, are expressed [8].

This disconnect raises an urgent question: how does social adversity interact with genetic predisposition to shape dementia outcomes? Understanding this interaction is not only scientifically relevant but also essential for designing equitable prevention strategies that recognize the dual impact of biology and environment, [9] and for ensuring that genetic risk information is interpreted appropriately across social strata [10]. Without this understanding, we risk implementing policies that exacerbate inequalities or misclassify risk in socially disadvantaged populations.

Two theoretical models offer competing predictions about how genes and environments interact to influence dementia risk. The social trigger model (Fig 1A) posits that genetic effects are magnified under high social adversity: stressful environments trigger the phenotypic expression of risky genes and amplify the benefits of protective variants [11]. In this model, genetic risk is masked in advantaged settings and revealed under strain [12]. By contrast, the social distinction model (Fig 1B) suggests

Institute for Health Research (NIHR, Ref: 198-1074). Funding has also been provided by the Economic and Social Research Council (ESRC). José M. Aravena was supported by a Fulbright Fellowship, a National Research and Development Agency of Chile (ANID) Fellowship, the Yale Social and Behavioral Sciences Research Fund, and grants from the Yale University Council on Latin American and Iberian Studies, and the Yale MacMillan Center for International and Area Studies. Xi Chen was supported by the National Institute on Aging grants R01AG077529 and P30AG021342. Becca R. Levy was supported by the National Institute on Aging grants R01AG067533 and U01AG032284.

**Competing interests:** The authors have declared that no competing interests exist.

that genetic effects are more pronounced among individuals with social advantage [13]. Here, supportive environments provide the physiological and psychological conditions that allow genes (whether protective or harmful) to fully manifest, while adversity exerts such a strong influence on health that it overshadows genetic variation [14].

Despite the importance of these competing models, few empirical studies have directly compared them in relation to dementia risk, particularly using multidimensional measures of social adversity and large-scale population data. Moreover, it remains unclear whether APOE-associated dementia risk is shaped or suppressed by the social environments in which individuals age [15].

To address this gap, we used harmonized data from two nationally representative studies of older adults with similar genetic ancestry and measurement protocols: the Health and Retirement Study (HRS) in the United States and the English Longitudinal Study of Ageing (ELSA). These studies offer rich data on APOE genotypes, social determinants of health, and longitudinal cognitive outcomes, enabling us with the unique opportunity to test gene-environment interactions at scale.

We hypothesized that data would support the social distinction model, for two reasons. First, previous studies of dementia risk factors that have found that as social adversity increases, so does the number of dementia risk factors [16]. Second, the health burden combined with the psychological stressors of living under social adversity can be substantial, negatively affecting epigenetic aging [17] and thus likely exceeding the protective or harmful effect of the APOE- ε2 and ε4 alleles, respectively. Conversely, among those living with low social adversity, psychological and health factors will be optimal leading to reduced impact on epigenetic aging. Therefore, the APOE alleles will have more room for their phenotypical expression, making evident its harmful or beneficial impact on dementia risk.

Accordingly, we tested two hypotheses:

a) The impact of APOE alleles on dementia risk will be more pronounced among individuals with low social adversity as opposed to those exposed to high social adversity.

b) Higher social adversity will be associated with elevated dementia risk, regardless of APOE genotype.

## Materials and methods

### Study design and population

Study report follows the STROBE statement for cohort studies (S1 File). This was a secondary data analysis of two prospective nationally representative sister cohort studies: the Health and Retirement Study (HRS) [18] and the English Longitudinal Study of Ageing (ELSA) [19]. HRS is a nationally representative study of more than 37,000 individuals 51 years or older in the U.S., that measures several economic, psychosocial, and health factors of aging. In 2006, the study added genetic information. HRS has followed people for 15 waves every two years from

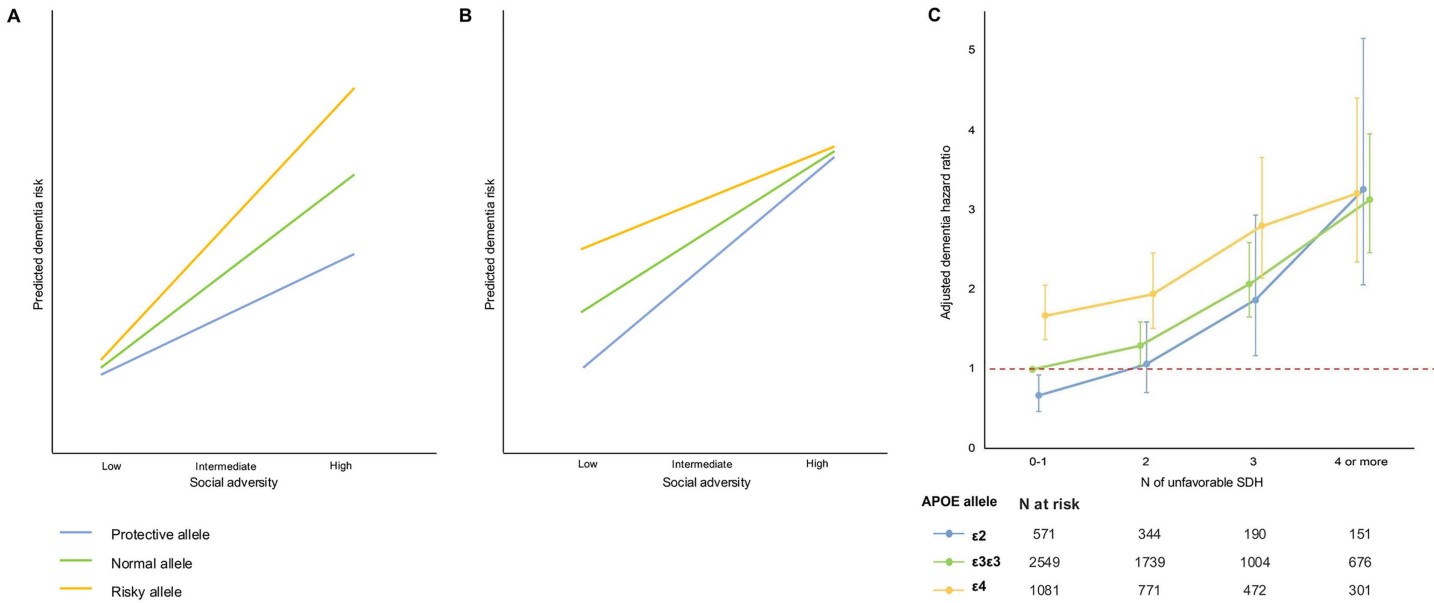

**Fig 1. Theoretical Gene-Environmental Models and Actual Results with HRS and ELSA Participants Illustrating Social Adversity and APOE Allele Interactions to Determine Risk of Developing Dementia.** Figures illustrating the Social Trigger Theoretical Model (A) and the Social Distinction Theoretical Model (B) for explaining the gene-environment interaction on cognitive health. Figure C describe the dementia Hazard Ratio by APOE allele profile and number of unfavorable social determinants of health (SDH) exposure compared to people at intermediate genetic risk (APOE-ε3ε3) exposed at 0-1 unfavorable SDH. APOE-ε2: ε2ε2, ε2ε3; APOE-ε4: ε2ε4, ε3ε4, ε4ε4. Figure C was designed based on our analyses performed with the pooled HRS and ELSA samples. N of unfavorable SDH exposed: less than upper secondary education, being in the 25th lower family income range, 25th lower score in neighborhood physical disorder, 25th lower score in neighborhood social cohesion, having experienced poorer health service from a doctor or hospital, not having some private healthcare insurance, 25th lower score in perceived social support, having experienced two or more different types of discrimination.

1992−93–2020−22. For this study, people were followed from the first HRS wave were genetic and SDH information was available, 2006 (wave 8) to 2018 (wave 14).

ELSA is a nationally representative study with more than 18,000 individuals in England. ELSA is the sister study of HRS applying almost the same measurement procedures but in a population 50 years or older. The study has followed people for 11 waves every two years from 2002−03–2018−20. ELSA added genetic information in 2004–2005. For this study, people were followed from the first ELSA wave were genetic and SDH information was available, 2010 (wave 5) to 2018 (wave 8).

## Inclusion criteria and selection of participants

For this analysis, we included all the participants ≥ 55 years, free of dementia at baseline, with available genetic (APOE) and SDH information at baseline, and at least two consecutive waves of the outcome (dementia). People with one or more wave gaps in dementia were excluded to avoid uncertainty about when dementia started.

Supporting information Figure S1 in S1 File describes the participants' study selection. At baseline, 7,954 and 5,531 participants had genetic and complete SDH information in HRS and ELSA, respectively. Of the HRS participants, 424 were excluded for not having cognitive or dementia information, 1,502 were less than 55 years old, 164 had dementia at baseline, and 67 had no follow-up. Of the ELSA participants, 15 were excluded for not having cognitive or dementia information, 106 were less than 55 years old, 124 had dementia at baseline, and 48 had no follow-up. The final pooled sample for analysis was 9,849 participants (HRS = 5,797, ELSA = 4,052) and followed up for up to 12 years (HRS = 12 years, ELSA = 8 years).

## Genetic information and APOE risk classification

Genetic data management and information by cohort study is described in detail in the Supporting information in S1 File S2 section.

Genetic variants used for identifying APOE allele were rs7412 and rs429358. Individuals of both studies were classified according to their APOE risk profile as low risk (APOE-ε2ε2 or ε2ε3 carriers), intermediate risk (APOE-ε3ε3 carriers), or high risk (APOE-ε2ε4, ε3ε4, or ε4ε4 carriers).

For sensitivity analyses, a polygenic risk score (PGS) for AD dementia that did not include the APOE allele was used. Participants were classified according to their PGS for AD dementia score by cohort and ancestry in three groups: low risk (< 25th quartile score), intermediate risk (25th - 75th quartile score), and high risk (> 75th quartile score) of dementia.

## Social adversity assessment

To create an index that reflect a theoretical explanation of cumulative health inequalities and considers the comprehensive definition of social adversity, we created an indicator using SDH measured in the same fashion in both cohort studies. These SDH reflected the five SDH areas defined by *Healthy People 2030 framework*: [20] education access, economic stability, neighborhood environment, healthcare access and quality, and social context. Specific SDH measured are described in the Supporting information S2 in S1 File section.

Social adversity was measured as the number of unfavorable social determinants (number of SDH in the lowest level or lower quartile). Unfavorable SDH considered were less than upper secondary education, low family income (<25th quartile income), high neighborhood physical disorder (<25th quartile score), low neighborhood social cohesion (<25th quartile score), having experienced healthcare discrimination, not having some private healthcare insurance, low social support (<25th quartile score), and having experienced two or more different types of discrimination. People were categorized into four levels of social adversity according to the number of unfavorable SDH presented: none or one unfavorable SDH (low social adversity), two, three, or four or more unfavorable SDH (high social adversity).

For sensitivity analysis, a second measure reflecting social advantage was created, counting the number of favorable social conditions (number of SDH in the highest level or highest quartile). Favorable SDH considered were tertiary education, high family income (>75th quartile income), low neighborhood physical disorder (>75th quartile score), high neighborhood social cohesion (>75th quartile score), no experiences of healthcare discrimination, having some private healthcare insurance, high mean social support (>75th quartile score), and no experiences of any type of discrimination. People were categorized into four levels of social advantage according to the number of favorable SDH presented: none or one favorable SDH (low social advantage), two, three, or four or more favorable SDH (high social advantage).

## Outcome variable – Dementia

The main outcome was dementia incidence, and it was identified in the same fashion in both cohort studies. Dementia was ascertained if the participant fulfilled one or more of the three criteria: a) self-reported diagnosis of AD or dementia by a physician; or b) the coexistence of cognitive and functional impairment, defined following a previously validated method in HRS and ELSA samples based on the Diagnostic and Statistical Manual of Mental Disorders fourth edition (DSM-IV), [21] as having at least two cognitive domain scores of 1.5 S.D. lower than the mean of the population stratified by education level for those cognitive domains. Cognitive domains measured were executive functions (HRS: backward counting, serial 7s; ELSA: verbal fluency, numeracy score), orientation (HRS and ELSA: date naming tasks), and memory (HRS and ELSA: delayed word recall, immediate word recall). Functional impairment was defined as having difficulties performing one or more activities of daily living (bathing, eating, dressing, getting into and out of bed, and walking across a room); or c) in the case of people unable to respond, a proxy informant diagnosis using the 16-item interview of cognitive functioning scale (IQCODE) was used [22]. IQCODE contrasts the present functional and cognitive performance with the past

two years' performance. A threshold of ≥ 3.38 has shown 0.82 sensitivity and 0.84 specificity to identify dementia, and no differences in performance between HRS and ELSA samples [23].

## Covariates

Covariates included in all the models were age, gender, and race or ethnicity. These variables were selected as those that predicted the exposure and the outcome and are not mediators in estimating the total effect.

Considering that a potential mediator of the association between social adversity and dementia risk is the number of dementia risk factors, [24] we measured for sensitivity analysis the presence (Yes/No) of eight evidence-based potentially modifiable risk factors of dementia: [7] hypertension, diabetes, obesity, smoking, high-frequency drinking (drinking more than 5 days a week), physical inactivity (participating <1 time per week in moderate or vigorous physical activity), self-reported hearing problems, and depression (CES-D score >2 points).

## Statistical analysis

For the analyses, we pooled participants from both studies to power the gene-environment interaction analyses that require a larger sample size. We used the Moore, Jacobson, & Fingerling power and sample size calculations formula for testing gene-environment interactions, [25] with power calculations based on a likelihood ratio test framework implemented in an open-source R package 'genpwr' (https://cran.r-project.org/web/packages/genpwr/index.html). Considering APOE is a dominant gene with prevalence of 12% for APOE-ε2 and 26% for ε4, and accounting that 12% of our population is exposed to high social vulnerability (≥ 4 unfavorable SDH), we estimated that a pooled sample size of 9,500 participants will have 86.0% and 94.2% power to detect APOE-ε2 and ε4-social vulnerability interactions of ratio 1.25 (Cohen's d effect size = 0.123, small effect size) in a $5.0 \times 10^{-5}$ significance level, respectively.

In the pooled sample, we described the baseline characteristics of included participants according to their APOE risk profile (low, intermediate, or high risk). Differences among groups in categorical and continuous variables were calculated using the Chi-Square test ($X^2$) and ANOVA test, respectively. The same procedure was conducted to compare cohorts' baseline characteristics and included and excluded participants' characteristics.

For all the main analyses, we used generalized Cox regression models considering a 95% confidence interval (CI) and two-sided p-value <0.05 for estimating the Hazard Ratio (HR) of dementia. We used random effect models ($I^2$) to estimate the overall effect in the pooled sample, [26] adjusting for age, sex, race, and country (the U.S. or England). Survival time was defined as the month of the interview when dementia was identified in the HRS or ELSA. Participants were censored if they were lost to follow-up, if they died without dementia before the end of the study, or if they completed the study with no dementia. All the model's hazard proportionality assumptions were assessed using the Schoenfeld residuals test.

To test our first hypothesis, we assessed the interaction effect between participants' APOE risk profile, and social adversity levels on dementia HR. Then, to examine if the impact of APOE alleles will be evident for those who are at low social adversity (≤ 1 unfavorable SDH) as opposed to those who are at high social adversity (≥ 4 unfavorable SDH), we estimated the HR of dementia by APOE risk profile in every social adversity level, using people at intermediate APOE risk (APOE-ε3ε3) exposed to the lowest level of social adversity (≤ 1 unfavorable SDH) as the reference value.

To test our second hypothesis, we estimated the HR of dementia by social adversity level in the complete pooled sample and by APOE risk profile, using the lowest level of social adversity (≤ 1 unfavorable SDH) as reference value. A p-value for assessing linear trends in every model was estimated. All these analyses were also performed by cohort (HRS and ELSA).

Six sensitivity analyses were conducted to assess the robustness of the findings and are described in detail in the Supporting information SI in S1 File 1 section. The analyses evaluated if the results remind consistent using social advantage as exposure, if the effect is maintained using a polygenic risk score excluding APOE allele, if potential reverse causality influenced results by excluding people with MCI at baseline, using a unique outcome assessment method based

on cognitive scores, adjusting by eight evidence-based dementia risk factors, and assessing potential selection bias by calculating the inverse probability of being selected in the study.

Finally, subgroup analyses by sex (female and male), race (Black and White people), and specific social determinants were conducted to explore group differences and to identify which social determinants showed the strongest interaction with APOE genotype in terms of increasing dementia risk. All the analyses were conducted using Python 3.8.5 'pandas' package (https://pandas.pydata.org/) and R version 4.3.3. packages 'survminer' (https://cran.r-project.org/web/packages/survminer/index.html), 'survival' (https://cran.r-project.org/web/packages/survival/index.html), and 'coxme' (https://cran.r-project.org/web/packages/coxme/index.html).

### Ethics statement

This study is a secondary analysis of publicly available, de-identified data from both cohort studies.

The ELSA study received ethical approval from the London Multi-Centre Research Ethics Committee, the National Hospital for Neurology and Neurosurgery & Institute of Neurology Joint Research Ethics Committee, and the South Central – Berkshire Research Ethics Committee. The HRS study was approved by the University of Michigan Institutional Review Board.

All participants in both studies provided informed consent, and all data used in this analysis were anonymized prior to access. No additional ethical approval was required for this secondary analysis.

## Results

Baseline characteristics and differences by APOE allele profile and cohort are described in Table 1 and Supporting information S1 Table in S1 File, respectively.

The APOE allele distribution in the population was 12.8% APOE-ε2 (ELSA = 13.5%, HRS = 12.2%), 60.6% APOE-ε3ε3 (ELSA = 59.5%, HRS = 61.4%), and 26.6% APOE-ε4 (ELSA = 27.0%, HRS = 26.4%), with no significant differences between samples (P-value = 0.077). People who are APOE-ε4 carriers were younger (mean age APOE-ε2 = 68.64, APOE-ε3ε3=68.68, APOE-ε4 = 67.82, P-value<0.001), a greater number of non-Hispanic Black people were APOE-ε4 carriers (APOE-ε2 = 8.3%, APOE-ε3ε3=6.6%, APOE-ε4 = 10.2%, P-value<0.001), Hispanic people were APOE-ε3ε3 carriers more often (APOE-ε2 = 4.2%, APOE-ε3ε3=9.0%, APOE-ε4 = 6.8%, P-value<0.001), and a greater proportion of APOE-ε2 carriers referred high social support (APOE-ε2 = 29.3%, APOE-ε3ε3=25,7%, APOE-ε4 = 23.9%, P-value = 0.009). No other differences were found among APOE allele groups at baseline. All these variables were included as covariates in the main and sensitivity analyses models.

### Social adversity-by-APOE allele risk interactions

As predicted by the first hypothesis, the impact of APOE alleles was more evident in people privileged to live with low social adversity as opposed to those exposed to greater social adversity for whom the impact of the APOE alleles had no significant impact (as suggested by the social distinction model); we found the interaction effect between social adversity and APOE allele profile was significant (P for social disadvantage x APOE interaction = 0.001) (Fig 1C).

In the low social adversity level (≤ 1 unfavorable SDH), people at low (ε2) and high (ε4) genetic risk profiles presented the lowest (APOE-ε2 HR = 0.67, 95%CI = 0.48–0.93) and the highest (APOE-ε4 HR = 1.68, 95%CI = 1.37–2.06) risk of dementia, respectively, compared to people at intermediate genetic risk (ε3ε3) exposed to ≤ 1 unfavorable SDH (Fig 1C, Table 2). In contrast, as social adversity levels increased, dementia risk was greater and similar among APOE allele profiles, compared to people at intermediate genetic risk (ε3ε3) exposed to ≤ 1 unfavorable SDH (≥ 4 unfavorable SDH, APOE-ε2 HR = 3.26, 95%CI = 2.06–5.16; APOE-ε3ε3 HR = 3.12, 95%CI = 2.47–3.95; APOE-ε4 HR = 3.21, 95%CI = 2.34–4.41, Random effect ($I^2$) =0.0%) (Fig 1C, Table 2).

**Table 1. Descriptive Statistics of the Sample at Baseline by APOE Allele Risk Profile.**

| | Total sample n: 9849 | Low risk (APOE-ε2ε2, ε2ε3) n: 1256 | Intermediate risk (APOE-ε3ε3) n: 5968 | High risk (APOE-ε2ε4, ε3ε4, ε4ε4) n: 2625 | P-value |
|---|---|---|---|---|---|
| Follow-up duration in years Mean, SD (IQR) | 6.88, 3.792 (4.0–10.0) | 6.94, 3.867 (4.0–12.0) | 6.90, 3.817 (4.0–10.0) | 6.79, 3.702 (4.0–10.0) | 0.371 |
| Age Mean, SD (IQR) | 68.45, 8.783 (62.0–74.0) | 68.64, 8.922 (62.0–75.0) | 68.68, 8.852 (62.0–75.0) | 67.82, 8.527 (61.0–73.0) | <0.001 |
| Sex n (%) | | | | | |
| Male | 4404 (44.7) | 554 (44.1) | 2689 (45.1) | 1161 (44.2) | 0.698 |
| Female | 5445 (55.3) | 702 (55.9) | 3279 (54.9) | 1464 (55.8) | |
| Race/ethnicity, n (%) | | | | | |
| Non-Hispanic White | 8832 (89.7) | 1132 (90.1) | 5406 (90.6) | 2294 (87.4) | <0.001 |
| Non-Hispanic Black | 766 (7.8) | 104 (8.3) | 394 (6.6) | 268 (10.2) | |
| Other (American Indian, Alaskan Native, Asian, and Pacific Islander) | 251 (2.5) | 20 (1.6) | 168 (2.8) | 63 (2.4) | |
| Hispanic | 454 (4.6) | 30 (4.2) | 320 (9.0) | 104 (6.8) | |
| Dementia risk factors, n (%) | | | | | |
| Diabetes | 1477 (15.0) | 185 (14.7) | 921 (15.4) | 371 (14.1) | 0.287 |
| Hypertension | 4954 (50.3) | 616 (49.0) | 3008 (50.4) | 1330 (50.7) | 0.619 |
| Obesity | 2959 (31.5) | 387 (32.2) | 1825 (32.0) | 747 (29.8) | 0.122 |
| High-frequency drinking | 1274 (13.0) | 166 (13.3) | 770 (13.0) | 338 (13.0) | 0.947 |
| Physical inactivity | 2252 (22.9) | 291 (23.2) | 1378 (23.1) | 583 (22.2) | 0.652 |
| Smoking | 1150 (11.7) | 150 (12.0) | 680 (11.4) | 320 (12.2) | 0.536 |
| Hearing problems | 435 (4.4) | 53 (4.2) | 273 (4.6) | 109 (4.2) | 0.635 |
| Depression | 1836 (18.6) | 231 (18.4) | 1127 (18.9) | 478 (18.2) | 0.729 |
| N of dementia risk factors, n (%) | | | | | |
| 0 | 1797 (18.2) | 232 (18.5) | 1086 (18.2) | 479 (18.2) | 0.977 |
| 1 | 3131 (31.8) | 395 (31.4) | 1888 (31.6) | 848 (32.3) | |
| 2 | 2512 (25.5) | 323 (25.7) | 1517 (25.4) | 672 (25.6) | |
| ≥3 | 2409 (24.5) | 306 (24.3) | 1475 (24.7) | 626 (23.8) | |
| Social determinants of health | | | | | |
| Education level, n (%) | | | | | |
| Less than upper secondary | 2066 (21.0) | 247 (19.7) | 1236 (20.7) | 583 (22.2) | 0.209 |
| Upper secondary and vocational | 5571 (56.6) | 704 (56.1) | 3398 (56.9) | 1469 (56.0) | |
| Tertiary | 2212 (22.5) | 305 (24.3) | 1334 (22.4) | 573 (21.8) | |
| Family income quartile by cohort, n (%) | | | | | |
| <25th quartile income | 2132 (21.6) | 268 (21.3) | 1296 (21.7) | 568 (21.6) | 0.998 |
| 25th – 75th quartile income | 5120 (52.0) | 656 (52.2) | 3102 (52.0) | 1362 (51.9) | |
| >75th quartile income | 2597 (26.4) | 332 (26.4) | 1570 (26.3) | 695 (26.5) | |
| Healthcare access, n (%) | | | | | |
| No private health insurance | 4776 (48.5) | 617 (49.1) | 2932 (49.1) | 1227 (46.7) | 0.112 |
| Healthcare discrimination | 546 (5.5) | 67 (5.3) | 331 (5.5) | 148 (5.6) | 0.928 |
| Neighborhood environment, n (%) | | | | | |
| Physical disorder | | | | | |
| Low (<25th) | 2071 (21.0) | 248 (19.7) | 1242 (20.8) | 581 (22.1) | 0.331 |
| Moderate (25th – 75th) | 4614 (46.8) | 585 (46.6) | 2800 (46.9) | 1229 (46.9) | |
| High (>75th) | 3164 (32.2) | 423 (33.7) | 1926 (32.3) | 815 (31.0) | |

*(Continued)*

**Table 1.** (Continued)

| | Total sample n: 9849 | Low risk (APOE-ε2ε2, ε2ε3) n: 1256 | Intermediate risk (APOE-ε3ε3) n: 5968 | High risk (APOE-ε2ε4, ε3ε4, ε4ε4) n: 2625 | P-value |
|---|---|---|---|---|---|
| Social cohesion difficulties | | | | | |
| Low (<25th) | 2079 (21.1) | 267 (21.3) | 1232 (20.6) | 580 (22.1) | 0.247 |
| Moderate (25th − 75th) | 4640 (47.1) | 568 (45.2) | 2829 (47.4) | 1243 (47.4) | |
| High (>75th) | 3130 (31.8) | 421 (33.5) | 1907 (32.0) | 802 (30.6) | |
| Social and community context, n (%) | | | | | |
| Social support | | | | | |
| Low (<25th) | 2221 (22.6) | 259 (20.6) | 1357 (22.7) | 605 (23.0) | 0.009 |
| Moderate (25th − 75th) | 5097 (51.8) | 629 (50.1) | 3076 (51.5) | 1392 (53.0) | |
| High (>75th) | 2531 (25.7) | 368 (29.3) | 1535 (25.7) | 628 (23.9) | |
| N of types of discrimination | | | | | |
| 0 | 4692 (47.6) | 605 (48.2) | 2850 (47.8) | 1237 (47.1) | 0.791 |
| 1 | 3358 (34.1) | 432 (34.4) | 2012 (33.7) | 914 (34.8) | |
| ≥2 | 1799 (18.3) | 219 (17.4) | 1106 (18.5) | 474 (18.1) | |
| Social adversity: n of unfavorable SDH | | | | | |
| ≤ 1 | 4201 (42.7) | 571 (45.5) | 2549 (42.7) | 1081 (41.2) | 0.143 |
| 2 | 2854 (29.0) | 344 (27.4) | 1739 (29.1) | 771 (29.4) | |
| 3 | 1666 (16.9) | 190 (15.1) | 1004 (16.8) | 472 (18.0) | |
| ≥ 4 | 1128 (11.4) | 151 (12.0) | 676 (11.3) | 301 (11.5) | |
| Any functional limitations in ADL, n (%) | 1340 (13.6) | 169 (13.5) | 826 (13.8) | 345 (13.1) | 0.676 |

SDH: Social Determinants of Health; IQR: interquartile range; ADL: Activities of daily living.

<25th: less than 25th quartile by cohort; 25th − 75th: between 25th and 75th quartile by cohort;>75th: more than 75th quartile by cohort.

Social adversity measured as N of unfavorable social determinants of health (SDH) exposed: Less than upper secondary education, being in the 25th lower family income range, 25th lower score in neighborhood physical disorder, 25th lower score in neighborhood social cohesion, having experienced poorer health service from a doctor or hospital, not having some private healthcare insurance, 25th lower score in perceived social support, having experienced two or more different types of discrimination.

P-value for chi-square test ($X^2$) for nominal and ordinal variables and ANOVA test for continuous variables comparison.

## Social adversity and the risk of dementia

In support of the second hypothesis, we found that greater social adversity was associated with a significantly higher risk of dementia (P for linear trend<0.001) (Fig 2A-2C). In the linear association, for every increment in social adversity levels, the risk of dementia increased 0.42 times (HR = 1.42, 95%CI = 1.28–1.56, $I^2 = 0.0\%$).

Regarding social adversity effect by APOE alleles groups, we found that greater social adversity was related to a higher risk of dementia in all the APOE allele profiles (Fig 3A-3C and Table 3).

To illustrate the impact of social adversity on APOE allele effect, compared to people at intermediate genetic risk (ε3ε3) exposed to ≤ 1 unfavorable SDH, people at high genetic risk (ε4) in the lowest level of social adversity presented a lower risk of dementia than people at low genetic risk (ε2) in the highest level of social adversity (APOE-ε4 and low social adversity HR = 1.68, 95%CI = 1.37–2.06; APOE-ε2 and high social adversity HR = 3.26, 95%CI = 2.06–5.16) (Fig 1C, Table 2).

## Social adversity, APOE allele, and dementia risk by country

In the assessment of results by study country (the U.S. and England), a trend supporting hypothesis one was observed, suggesting that the pooled sample with the larger power is needed to detect the gene-environment interaction effect. Differences in dementia risk were found between APOE allele profiles in the lowest social adversity level, although these

*(Continued)*

**Table 2. Interaction Between Social Adversity Level and APOE Allele Risk Profile on Greater Dementia Risk.**

| APOE allele risk profile | N of unfavorable SDH | ELSA | | HRS | | Pooled analysis with ELSA and HRS |
|---|---|---|---|---|---|---|
| | | N total/ N events | HR (95%CI) | N total/ N events | HR (95%CI) | HR (95%CI) |
| Low risk (ε2ε2, ε2ε3) | ≤ 1 SDH | 168/2 | 0.28 (0.07–1.18) | 403/37 | 0.72 (0.51–1.02) | 0.67 (0.48–0.93)** |
| | 2 SDH | 186/8 | 1.16 (0.53–2.56) | 158/18 | 0.99 (0.61–1.61) | 1.06 (0.71–1.59) |
| | 3 SDH | 113/8 | 2.01 (0.91–4.43) | 77/12 | 1.67 (0.93–2.98) | 1.85 (1.17–2.93)** |
| | ≥ 4 SDH | 82/4 | 1.77 (0.62–5.07) | 69/16 | 3.78 (2.26–6.31)*** | 3.26 (2.06–5.16)*** |
| Intermediate risk (ε3ε3) | ≤ 1 SDH | 704/28 | Ref | 1845/222 | Ref | Ref |
| | 2 SDH | 878/36 | 1.08 (0.65–1.78) | 861/118 | 1.33 (1.06–1.66)** | 1.30 (1.06–1.59)** |
| | 3 SDH | 527/28 | 1.85 (1.09–3.15)* | 477/89 | 2.10 (1.64–2.70)*** | 2.07 (1.66–2.59)*** |
| | ≥ 4 SDH | 301/23 | 3.26 (1.87–5.71)*** | 375/82 | 3.03 (2.34–3.94)*** | 3.12 (2.47–3.95)*** |
| High risk (ε2ε4, ε3ε4, ε4ε4) | ≤ 1 SDH | 319/16 | 1.36 (0.73–2.52) | 762/132 | 1.71 (1.38–2.12)*** | 1.68 (1.37–2.06)*** |
| | 2 SDH | 401/22 | 1.77 (1.01–3.11)* | 370/67 | 1.95 (1.48–2.57)*** | 1.93 (1.51–2.46)*** |
| | 3 SDH | 231/16 | 2.10 (1.13–3.90)** | 241/54 | 2.99 (2.21–4.04)*** | 2.80 (2.14–3.66)*** |
| | ≥ 4 SDH | 142/15 | 4.09 (2.17–7.71)*** | 159/33 | 2.88 (1.98–4.19)*** | 3.21 (2.34–4.41)*** |

HRS: the Health and Retirement Study; ELSA: the English Longitudinal Study on Aging; SDH: Social Determinants of Health; HR: Hazard ratio; CI: Confidence Interval; Ref: reference value.

*: p<0.05; **: p<0.01; ***: p<0.001.

All models were adjusted by age, sex, and race. The pooled model was additionally adjusted by cohort (HRS or ELSA). Random effect ($I^2$) for the pooled model: 0.0%.

Social adversity measured as N of unfavorable SDH exposed: less than upper secondary education, being in the 25th lower family income range, 25th lower score in neighborhood physical disorder, 25th lower score in neighborhood social cohesion, having experienced poorer health service from a doctor or hospital, not having some private healthcare insurance, 25th lower score in perceived social support, having experienced two or more different types of discrimination.

Model properties: all the models' long-rank test p-value<0.001. Fully adjusted model hazard proportionality assumption based on Schoenfeld residuals. The models meet the hazard proportionality assumption (p-value >0.05). The variables meet the hazard proportionality assumption (p-value >0.05).

were non-significant (Table 2). Consistent with the hypothesis, as social adversity levels increased, the risk of dementia was similar for individuals with intermediate and high genetic risk in the ELSA and HRS samples (Table 2).

In both samples, hypothesis two was supported, with consistent patterns across both cohorts indicating that greater social adversity increased the risk of dementia in all the APOE allele groups (Fig 2B and 2C, Table 3).

## Sensitivity analyses

First, to evaluate if the results remind consistent if we use social advantage level as exposure, we created an indicator of social advantage counting the number of favorable social determinants (e.g., high income, high neighborhood social cohesion, and high social support). Using this definition, the results follow the same pattern as the main analysis, where at low social advantage (≤ 1 favorable SDH), no statistically significant differences between APOE alleles were observed (APOE-ε2 HR=0.68, 95%CI=0.35–1.34; APOE-ε4 HR=1.34, 95%CI=0.90–2.01), while at the greater level of social advantage (≥ 4 favorable SDH), the differences between APOE-ε2 and APOE-ε4 were statistically significant in the expected direction (APOE-ε2 HR=0.29, 95%CI=0.19–0.43; APOE-ε3ε3 HR=0.47, 95%CI=0.35–0.62; APOE-ε4 HR=0.75, 95%CI=0.55–1.01) (S2 Fig and S2 Table in S1 File).

Second, to determine if the social adversity effect is maintained using a different definition of the dementia genetic risk profile, individuals were categorized into three genetic risk profiles based on quartiles of AD dementia polygenic

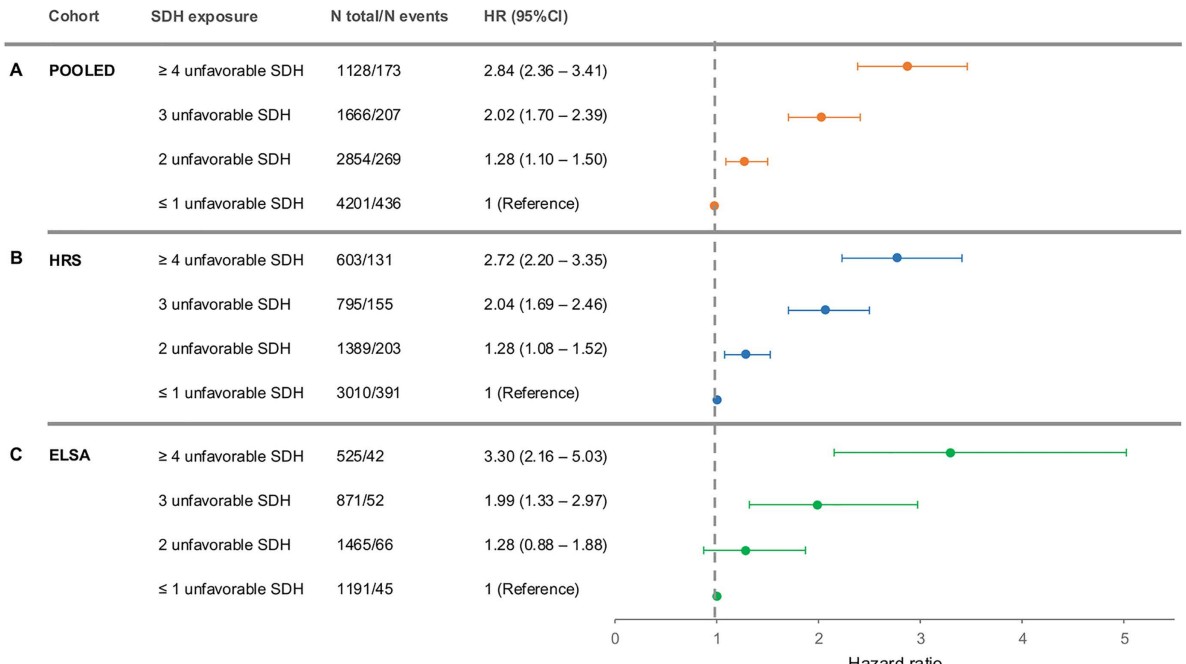

**Fig 2. Association Between Social Adversity Level and Greater Risk of Dementia.** Forest-plot for fully adjusted dementia Hazard ratio (HR) by exposure to number of unfavorable social determinants of health (SDH) (social adversity) in the population by pooled sample **(A)**, the **U.**S. cohort and **(B)**, the England cohort **(C)**. Unfavorable SDH exposed: less than upper secondary education, being in the 25th lower family income range, 25th lower score in neighborhood physical disorder, 25th lower score in neighborhood social cohesion, having experienced poorer health service from a doctor or hospital, not having some private healthcare insurance, 25th lower score in perceived social support, having experienced two or more different types of discrimination.

risk scores that do not include APOE alleles: low (< 25th quartile score), intermediate (25th - 75th quartile score), and high (> 75th quartile score) polygenic risk profiles. Results indicated that the impact of social adversity on increasing dementia risk remained similar across different polygenic risk profiles, as observed in the main analyses (S3 Table in S1 File).

Thirdly, to address the possibility of reverse causality, we excluded people with mild cognitive impairment (n = 494) or dementia at baseline. The results obtained were similar to the main analyses, confirming a low possibility of reverse causality in our findings (S4 Table in S1 File).

Fourth, to assess whether the results are influenced by the outcome assessment method, we included cases of dementia based solely on cognitive scores and functional limitations, finding that the results followed the same direction that those obtained in the main analyses (S5 Table in S1 File).

Fifth, to examine if results remain consistent after considering dementia risk factor burden, a mediator in the causal path between social adversity and dementia risk, we conducted the same analyses but additionally adjusting by eight evidence-based potentially modifiable risk factors of dementia [7]. As expected, the association effect' sizes were slightly reduced, but all the analyses presented the same direction and statistically significant association as the main analysis (S6 Table in S1 File).

Finally, to address potential selection bias due to differences between included and excluded individuals, we calculated the inverse probability of being selected by study cohort [27]. All analyses were weighted by the inverse probability of being selected for the study. Observed results from the weighted analysis did not differ from those obtained in the primary analyses, confirming a low chance of selection bias in our findings (S7 and S8 Tables in S1 File).

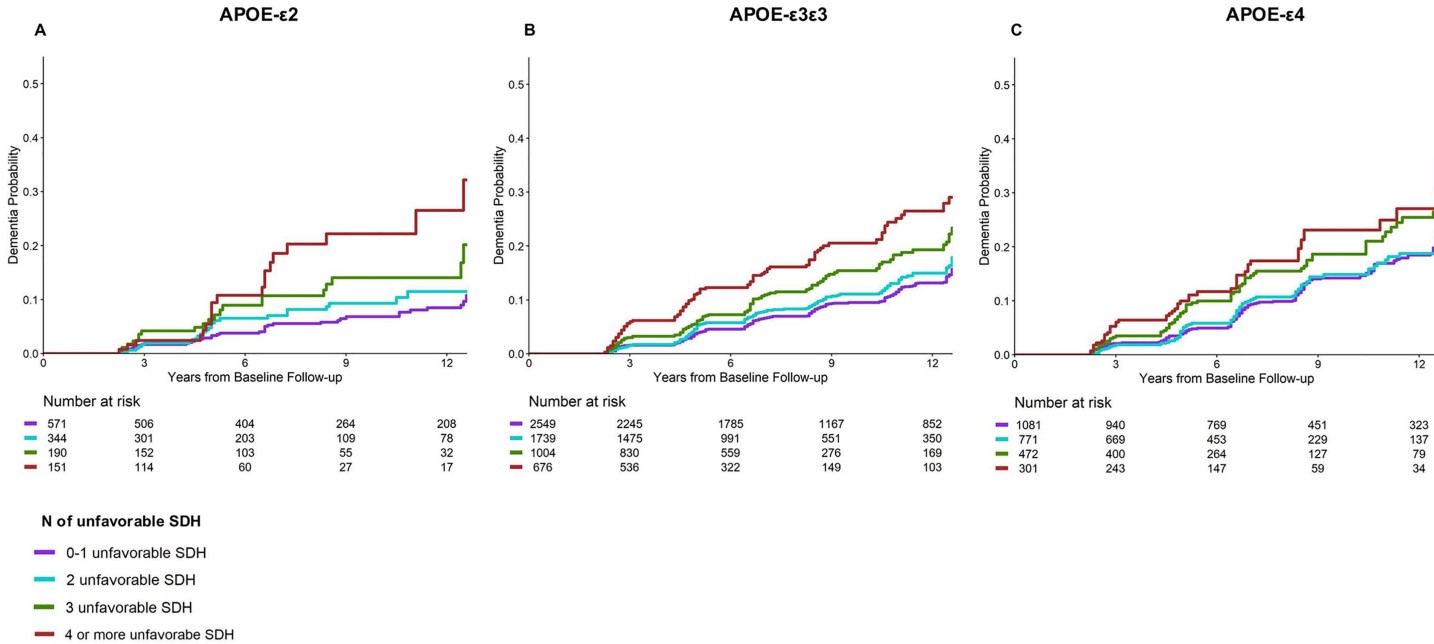

**Fig 3. Cumulative Incidence of Dementia and Social Adversity Level by APOE Risk Profile.** Unadjusted Kaplan-Meier survival curve for dementia incidence in people exposed to levels of social adversity (number of unfavorable social determinants of health -SDH-) in the pooled sample at low genetic risk (APOE-ε2ε2, ε2ε3)(A), intermediate genetic risk (APOE-ε3ε3)(B), and high genetic risk (APOE-ε2ε4, ε3ε4, ε4ε4)(C). N of unfavorable SDH exposed: less than upper secondary education, being in the 25th lower family income range, 25th lower score in neighborhood physical disorder, 25th lower score in neighborhood social cohesion, having experienced poorer health service from a doctor or hospital, not having some private healthcare insurance, 25th lower score in perceived social support, having experienced two or more different types of discrimination.

## Sub-group analyses

Subgroup analysis by sex showed that, in both males and females, the same pattern of APOE allele impact was observed: the effect was more evident among individuals living with low social adversity, whereas among those exposed to greater social adversity, the influence of the APOE allele was less pronounced (S9.a Table in S1 File). However, the impact of social adversity on APOE genotype subgroups was more evident in women. We also observed that the effect of social adversity on male APOE-ε2 carriers was less pronounced than in female carriers, whereas the opposite pattern was found for APOE-ε4, with a stronger effect among female carriers compared to male carriers (S9.b Table in S1 File).

Regarding racial differences (Black and White individuals), the same effect was observed: people at high social disadvantage presented the greatest risk of dementia, regardless of APOE allele (S10.a Table in S1 File). Among White participants, APOE-ε2 carriers at social advantage presented a lower risk of dementia, and APOE-ε4 carriers at social advantage had a greater risk of dementia than APOE-ε3ε3 carriers at social advantage (S10.b Table in S1 File). In contrast, among Black participants, APOE-ε2 carriers at social advantage did not present a lower risk of dementia, and APOE-ε4 carriers at social advantage showed no difference in dementia risk compared to APOE-ε3ε3 carriers at social advantage (S10.b Table in S1 File).

## Specific social determinants and APOE allele groups

Regarding individual social determinants, having a lower education level and lower family income presented the strongest interactions and were associated with a greater risk of dementia, regardless of APOE allele group (S11.a Table in S1 File). Interestingly, among APOE-ε4 carriers, only having less than upper secondary education was associated with greater

**Table 3. Association Between Social Adversity Level and Greater Dementia Risk by APOE Allele Risk Profile.**

| APOE allele risk profile | N of unfavorable SDH | ELSA | | HRS | | Pooled analysis with ELSA and HRS |
|---|---|---|---|---|---|---|
| | | N at risk/ N event | HR (95%CI) | N at risk/ N event | HR (95%CI) | HR (95%CI) |
| **Low risk (ε2ε2, ε2ε3)** | ≤ 1 SDH | 168/2 | Ref | 403/37 | Ref | Ref |
| | 2 SDH | 186/8 | 4.08 (0.86–19.34) | 158/18 | 1.33 (0.75–2.35) | 1.55 (0.93–2.57) |
| | 3 SDH | 113/8 | 7.48 (1.58–35.34)** | 77/12 | 2.23 (1.15–4.30)** | 2.74 (1.58–4.76)*** |
| | ≥ 4 SDH | 82/4 | 6.37 (1.14–35.49)** | 69/16 | 4.75 (2.51–8.99)*** | 4.47 (2.52–7.94)*** |
| | p-for trend | 0.021 | | <0.001 | | <0.001 |
| **Intermediate risk (ε3ε3)** | ≤ 1 SDH | 704/28 | Ref | 1845/222 | Ref | Ref |
| | 2 SDH | 878/36 | 1.06 (0.64–1.75) | 861/118 | 1.31 (1.05–1.64)** | 1.27 (1.04–1.56)* |
| | 3 SDH | 527/28 | 1.79 (1.05–3.05)* | 477/89 | 2.08 (1.61–2.67)*** | 2.03 (1.62–2.55)*** |
| | ≥ 4 SDH | 301/23 | 3.23 (1.85–5.66)*** | 375/82 | 2.95 (2.26–3.85)*** | 3.03 (2.39–3.86)*** |
| | p-for trend | <0.001 | | <0.001 | | <0.001 |
| **High risk (ε2ε4, ε3ε4, ε4ε4)** | ≤ 1 SDH | 319/16 | Ref | 762/132 | Ref | Ref |
| | 2 SDH | 401/22 | 1.32 (0.69–2.52) | 370/67 | 1.18 (0.88–1.59) | 1.19 (0.91–1.55) |
| | 3 SDH | 231/16 | 1.58 (0.79–3.17) | 241/54 | 1.84 (1.33–2.54)*** | 1.76 (1.31–2.35)*** |
| | ≥ 4 SDH | 142/15 | 3.14 (1.55–6.40)*** | 159/33 | 1.86 (1.24–2.78)** | 2.09 (1.48–2.95)*** |
| | p-for trend | 0.001 | | <0.001 | | <0.001 |

HRS: the Health and Retirement Study; ELSA: the English Longitudinal Study on Aging; SDH: Social Determinants of Health; HR: Hazard ratio; CI: Confidence Interval; Ref: reference value; p-for trend: p-value for linear trend.

*: p<0.05; **: p<0.01; ***: p<0.001

All models were adjusted by age, sex, and race. The pooled models were additionally adjusted by cohort (HRS or ELSA). Random effects ($I^2$) for the three pooled models: 0.0%

Social adversity measured as N of unfavorable SDH exposed: less than upper secondary education, being in the 25th lower family income range, 25th lower score in neighborhood physical disorder, 25th lower score in neighborhood social cohesion, having experienced poorer health service from a doctor or hospital, not having some private healthcare insurance, 25th lower score in perceived social support, having experienced two or more different types of discrimination.

Model properties: all the models' long-rank test p-value<0.001. Fully adjusted model hazard proportionality assumption based on Schoenfeld residuals. The models meet the hazard proportionality assumption (p-value >0.05). The variables meet the hazard proportionality assumption (p-value >0.05).

dementia risk, whereas dementia risk remained elevated across all income levels (intermediate and low) when compared to those with high family income (S11.a Table in S1 File).

In terms of healthcare-related factors, lower access to and quality of healthcare were associated with increased dementia risk among APOE-ε3ε3 and APOE-ε4 carriers (S11.b Table in S1 File). Meanwhile, lower perceived social support and higher levels of neighborhood physical disorder were linked to greater dementia risk among APOE-ε2 and APOE-ε3ε3 carriers (S11.b and 11.c Tables in S1 File). Finally, lower neighborhood social cohesion was associated with increased dementia risk, but only among APOE-ε3ε3 carriers (S11.c Table in S1 File).

## Discussion

In this study, we addressed a central question in the gene-environment debate: how does social position interact with genetic predisposition to shape dementia risk? Our findings support the social distinction model, which posits that genetic effects are more pronounced in socially advantaged contexts. Specifically, we found that the influence of APOE alleles on dementia risk was most evident among individuals with low social adversity, while among those experiencing high social adversity, APOE genotype had minimal impact. These findings suggest that social context can either amplify or attenuate the phenotypic expression of genetic risk for cognitive decline associated with APOE.

Consistent with our second hypothesis, we found that greater cumulative social adversity was associated with higher dementia risk, regardless of APOE status. Strikingly, individuals with high genetic risk (APOE-ε4) but low social adversity had a lower risk of dementia than those with low genetic risk (APOE-ε2) but high social adversity. These results highlight the dominant role of social conditions in modulating biological vulnerability, a finding with profound implications for precision medicine and public health equity.

Our results build on prior work showing that socioeconomic deprivation is associated with increased risk of dementia and poorer brain functioning [28–30]. Unlike earlier studies, we demonstrate that all APOE allele variants, as well as polygenic risk for AD, interact with cumulative social adversity in a manner consistent with the social distinction model. This suggests that the social gradient, whether measured by adversity or advantage, modulates the influence of genetic risk on dementia outcomes. These findings raise important considerations for gene discovery efforts, such as genome-wide association studies (GWAS) and polygenic risk score (PRS) development: the expression and statistical detectability of genetic associations with cognitive aging may depend on the social context in which individuals age. Without accounting for social conditions as potential moderators, studies may overlook candidate genes or misestimate their significance across socially diverse populations [31].

In the same line, our findings underscore the need for future research to develop new methods for evaluating the accuracy and clinical utility of genetic risk models for dementia; approaches that move beyond treating social determinants as covariates or fixed risk factors and instead conceptualize them as dynamic modulators of gene expression. Taken together, these insights point to the importance of pursuing multi-level, gene-environment approaches to dementia prevention. Targeting either APOE-related pathways or social environments in isolation may be insufficient to reduce dementia risk, particularly among socially disadvantaged populations [9].

Among individuals exposed to high social adversity, dementia risk was not only similar across APOE genotypes but also higher than in any other genetic or social adversity group. This finding underscore how multidimensional social adversity can outweigh both protective and risky genetic profiles in determining cognitive health outcomes [14]. These results highlight the importance of developing upstream social and public health policies aimed at reducing social adversity as a strategy for dementia prevention that can benefit individuals regardless of their genetic risk. Furthermore, they support the integration of comprehensive measures of social adversity into precision medicine frameworks and dementia risk prediction models.

Nevertheless, we acknowledge that some individual SDH may interact with genetic risk in ways more consistent with the social trigger model, [12,32] and that interactions may vary across specific cognitive domains, other genetic variants, or life-course exposures [33]. Our findings also suggest that these interactions may differ across social identities, with variations by sex and race indicating that social context may not only amplify or buffer genetic risk, but do so in ways shaped by structural inequality. Future research should explore these intersectional effects more explicitly, integrating sex, race, and other axes of identity into gene-environment models to better capture how societal structures shape dementia risk. Continued research is needed to examine these complexities and better characterize how social environments shape gene-related dementia risk.

One plausible pathway by which social adversity may influence dementia is through modifiable lifestyles and chronic disease risk factors [34]. The burden of potentially modifiable dementia risk factors is significantly greater in populations who live in poorer regions and low- and middle-income countries [35]. Yet, studies suggest that the association between socioeconomic status and dementia risk is mediated only up to 25% by unhealthy lifestyles [36]. In our analysis, controlling for eight evidence-based risk factors of dementia attenuated effect sizes, but results remained consistent, suggesting that other relevant factors related to chronic psychological stress may also contribute to the association between social adversity and dementia risk. This likely includes situations frequently faced by older people at social adversity, such as food insecurity, [37] exposure to environmental contaminants, [38] traumatic experiences, [39] age discrimination, [40] and lack of appropriate healthcare access [41]. Future moderator and mediator analyses are vital to informing experimental studies testing structural interventions for dementia risk reduction.

In our study, social adversity displayed a similar pattern in HRS and ELSA of impact on dementia risk in both samples, reinforcing its association with brain health in both social contexts. This is impressive given the many differences between the U.S. and England, such as the amounts of income inequality, and neighborhood segregation, as well as the type of healthcare systems [42]. Nonetheless, we acknowledge that the American and British populations share many cultural and genetic ancestry features that can favor the similarity of results between samples. Results demonstrating that social factors may be more relevant than genetic ancestry on dementia risk have also been found in Latin American populations, [43] reinforcing the validity of our results. Despite this, we encourage the replication of these findings in additional places, particularly low—and middle-income countries.

A limitation of our study is that the two final datasets could have had larger racial and ethnic diversity. In future research studying the gene-environment models, examining the additional contribution of race, ethnicity, and exposure to SDH to the patterns described will be relevant. Another limitation is that SDH were measured at a single point in time. However, considering that the U.S. and the England have faced the most significant decrease in social mobility in the last 50 years, [44] it is likely that these variables stayed consistent for most of the participants. Finally, while this study identifies statistically robust gene-environment interactions, it does not establish a causal relationship between social adversity and the modulation of APOE-related dementia risk. Social adversity was not exogenously assigned, and residual confounding cannot be ruled out. Our findings should be interpreted as descriptive of differential risk patterns that are nonetheless critical for informing future research, prevention strategies, and equity-oriented risk prediction.

Our study has several strengths. We used two of the largest and most rigorously harmonized longitudinal cohorts of aging, HRS and ELSA, which enabled prospective analysis of dementia across diverse national settings. We applied a comprehensive, multidimensional operationalization of social determinants of health (SDH) based on the Healthy People 2030 framework, [20] capturing economic, educational, healthcare, social, and environmental dimensions. Unlike most prior studies, we included all APOE genotypes, allowing for nuanced analyses of gene–environment interactions. Dementia diagnoses were harmonized using validated algorithms across cohorts, enhancing comparability. Finally, we conducted six complementary sensitivity analyses, reinforcing the internal validity and robustness of our findings.

Beyond these methodological strengths, this study makes a unique contribution by being the first to robustly test the social distinction model of gene–environment interaction in the context of dementia. Unlike prior research that has focused on additive effects of genes and environment, our findings demonstrate how social advantage and disadvantage modify the expression of genetic risk, revealing that genetic risk may be less detectable, and potentially less actionable, in contexts of structural adversity. These insights help explain disparities in dementia incidence across social groups and suggest that the predictive value of genetic information is not universal, but conditional on societal context. This novel approach deepens our understanding of how macro-level factors shape biological vulnerability and offers a compelling case for incorporating SDH into precision brain health efforts.

In conclusion, our findings demonstrate that the influence of APOE genotype on dementia risk is shaped by social position, consistent with a social-distinction model. Genetic effects were amplified under social advantage and diminished under adversity. These results call for an integrated approach to dementia prevention that accounts for both biological and social risk, and that prioritizes structural strategies to reduce adversity, particularly among vulnerable groups. Equitable public health and precision medicine efforts will require a dual lens on genetic and social risk to maximize brain health in aging populations.

## Supporting information

**S1 File. Supporting information file.**
(DOCX)

## Acknowledgments

Authors would like to thanks to Dr Olesya Ajnakina for the assistance and support in managing the genetic data for ELSA study.

## Author contributions

**Conceptualization:** Jose M Aravena, Becca R. Levy.

**Data curation:** Jose M Aravena.

**Formal analysis:** Jose M Aravena, Xi Chen.

**Funding acquisition:** Jose M Aravena, Xi Chen, Becca R. Levy.

**Investigation:** Jose M Aravena, Becca R. Levy.

**Methodology:** Jose M Aravena, Xi Chen, Becca R. Levy.

**Project administration:** Jose M Aravena.

**Resources:** Jose M Aravena.

**Software:** Jose M Aravena.

**Supervision:** Jose M Aravena, Xi Chen, Becca R. Levy.

**Validation:** Jose M Aravena, Xi Chen.

**Visualization:** Jose M Aravena.

**Writing – original draft:** Jose M Aravena, Xi Chen, Becca R. Levy.

**Writing – review & editing:** Jose M Aravena, Xi Chen, Becca R. Levy.

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
