## [Decision Letter · Decision Letter 0]

29 Aug 2025

Dear Dr. Aravena,

Thank you for submitting your manuscript to PLOS ONE. After careful consideration, we feel that it has merit but does not fully meet PLOS ONE’s publication criteria as it currently stands. Therefore, we invite you to submit a revised version of the manuscript that addresses the points raised during the review process.

We look forward to receiving your revised manuscript.

Kind regards,

Ioannis Liampas, MD. PhD

Academic Editor

PLOS ONE

Journal Requirements:

 “The Health and Retirement Study (HRS) is supported by the National Institute on Aging (NIA U01AG009740) and the Social Security Administration funding the Health and Retirement Study. The English Longitudinal Study of Ageing (ELSA) is funded by the National Institute on Aging (Ref: R01AG017644) and by a consortium of UK government departments: Department for Health and Social Care; Department for Transport; Department for Work and Pensions, which is coordinated by the National Institute for Health Research (NIHR, Ref: 198-1074). Funding has also been provided by the Economic and Social Research Council (ESRC).

José M. Aravena was supported by a Fulbright Fellowship, a National Research and Development Agency of Chile (ANID) Fellowship, the Yale Social and Behavioral Sciences Research Fund, and grants from the Yale University Council on Latin American and Iberian Studies, and the Yale MacMillan Center for International and Area Studies. Xi Chen was supported by the National Institute on Aging grants R01AG077529 and P30AG021342. Becca R. Levy was supported by the National Institute on Aging grants R01AG067533 and U01AG032284.”

5. We note that you have indicated that there are restrictions to data sharing for this study. PLOS only allows data to be available upon request if there are legal or ethical restrictions on sharing data publicly. For more information on unacceptable data access restrictions, please see http://journals.plos.org/plosone/s/data-availability#loc-unacceptable-data-access-restrictions.

Reviewers' comments:

Reviewer's Responses to Questions

**Comments to the Author**

1. Is the manuscript technically sound, and do the data support the conclusions?

Reviewer #1: Yes

Reviewer #2: Yes

Reviewer #3: Yes

2. Has the statistical analysis been performed appropriately and rigorously?

Reviewer #1: Yes

Reviewer #2: Yes

Reviewer #3: I Don't Know

3. Have the authors made all data underlying the findings in their manuscript fully available?

Reviewer #1: Yes

Reviewer #2: Yes

Reviewer #3: Yes

4. Is the manuscript presented in an intelligible fashion and written in standard English?

Reviewer #1: Yes

Reviewer #2: Yes

Reviewer #3: Yes

Reviewer #1: The manuscript is well-written and effectively outlines the interaction between environmental and genetic factors as contributors to Alzheimer’s disease (AD) risk among a diverse cohort. The methodology is clearly described and appropriately cited, and the results are presented in a structured and coherent manner. The discussion thoughtfully interprets the findings and is supported by relevant literature. The following are a few suggestions to enhance the manuscript further:

1. While Table 1 presents descriptive statistics stratified by APOE allele risk profile, it would be helpful to also include a summary of participant characteristics for the full sample, regardless of APOE status. This would provide a clearer overall picture of the study population.

2. Author should also explore whether the interaction between APOE genotype and social adversity differs by sex or racial/ethnic group.

3. Since social adversity was constructed using five domains of social determinants of health (SDH), it would be informative to identify which specific domain(s) demonstrated the strongest interaction with APOE genotype in relation to dementia risk.

4. In addition to the limitations already outlined, add the unique strengths and contributions of this study.

Reviewer #2: The authors demonstrated that while genetic factors, particularly APOE status, are central to dementia risk prediction, their impact is strongly moderated by social position. Using data from two large longitudinal cohorts (HRS and ELSA), they constructed a social adversity index based on education, economic stability, healthcare, neighborhood, and social context. Their analyses revealed that genetic risk was most evident among individuals with social advantage, whereas high social adversity elevated dementia risk across all genotypes. Notably, individuals with high genetic risk but social advantage had a lower dementia risk than those with low genetic risk but high social adversity. These findings highlight that social environments substantially shape the effect of genetic risk, emphasizing the importance of addressing social adversity to reduce dementia risk and improve equity in prevention strategies. Overall, the authors did an excellent job presenting their study. The manuscript is well written, highly readable, and demonstrates careful scientific consideration. The integration of genetic and social determinants of health is both timely and impactful, making this a strong contribution to the dementia prevention literature.

Reviewer #3: The manuscript “Unequal Expression: Social Position Modulates APOE Genotype Risk of Dementia” demonstrates that genetic risks affect mostly those with social advantage, and those with high social adversity have higher risks for dementia regardless of their genetic risks. The manuscript is well-written, the aims are clear, and it’s well-presented with acceptable flow. The idea in this manuscript is novel, and the data adds knowledge to the field. I do not have much, but the following comment for the improvement of the manuscript: The resolution for figures 1-3 is very low.

**Do you want your identity to be public for this peer review?** For information about this choice, including consent withdrawal, please see our Privacy Policy

Reviewer #1: No

Reviewer #2: No

Reviewer #3: No

---

## [Author Response · Author response to Decision Letter 1]

7 Oct 2025

October 06th, 2025

Ioannis Liampas, MD, PhD

Academic Editor

PLOS ONE

RE: Manuscript ID PONE-D-25-31823 entitled: " Unequal Expression: Social Position Modulates APOE Genotype Risk of Dementia.”

Dear Dr. Liampas,

We appreciate the opportunity to address the editors and reviewers’ comments and revise our manuscript. We appreciate the chance to improve the formatting and results provided by our manuscript.

All page and paragraph numbers refer to locations in the revised manuscript.

As part of our response, we attach:

- Separate document uploaded as “Response to Reviewers” that addresses the issues raised in the below Executive Editor’s comments (with their helpful comments in bold and our responses not in bold).

- Revised Word document with track changes highlighting those changes in the manuscript itself entitled “Revised Manuscript with Track Changes.”

- Revised Word document without the changes highlighted (clean copy) entitled “Manuscript.”

Role of Funder Statement

Thank you again for your time and the opportunity to improve our manuscript.

Sincerely,

José M. Aravena, OT, MS, PhD

Department of Social & Behavioral Sciences

School of Public Health

Yale University.

---

## [Editor Report · Decision Letter 1]

16 Oct 2025

Unequal Expression: Social Position Modulates APOE Genotype Risk of Dementia.

PONE-D-25-31823R1

Dear Dr. Aravena,

We’re pleased to inform you that your manuscript has been judged scientifically suitable for publication and will be formally accepted for publication once it meets all outstanding technical requirements.

Kind regards,

Ioannis Liampas, MD. PhD

Academic Editor

PLOS ONE

---

## [Editor Report · Acceptance letter]

PONE-D-25-31823R1

PLOS ONE

Dear Dr. Aravena,

I'm pleased to inform you that your manuscript has been deemed suitable for publication in PLOS ONE. Congratulations! Your manuscript is now being handed over to our production team.

Kind regards,

on behalf of

Dr. Ioannis Liampas

Academic Editor

PLOS ONE